# Flexible large-area ultrasound arrays for medical applications made using embossed polymer structures

Paul L. M. J. van Neer[1,6], Laurens C. J. M. Peters[2,6], Roy G. F. A. Verbeek[2], Bart Peeters[2], Gerard de Haas[2], Lars Hörchens [1], Laurent Fillinger [1], Thijs Schrama[1], Egon J. W. Merks-Swolfs[1], Kaj Gijsbertse[3], Anne E. C. M. Saris[4], Moein Mozaffarzadeh[4], Jan M. Menssen[4], Chris L. de Korte [4,5], Jan-Laurens P. J. van der Steen[2], Arno W. F. Volker[1] & Gerwin H. Gelinck [2] ✉

With the huge progress in micro-electronics and artificial intelligence, the ultrasound probe has become the bottleneck in further adoption of ultrasound beyond the clinical setting (e.g. home and monitoring applications). Today, ultrasound transducers have a small aperture, are bulky, contain lead and are expensive to fabricate. Furthermore, they are rigid, which limits their integration into flexible skin patches. New ways to fabricate flexible ultrasound patches have therefore attracted much attention recently. First prototypes typically use the same lead-containing piezo-electric materials, and are made using micro-assembly of rigid active components on plastic or rubber-like substrates. We present an ultrasound transducer-on-foil technology based on thermal embossing of a piezoelectric polymer. High-quality two-dimensional ultrasound images of a tissue mimicking phantom are obtained. Mechanical flexibility and effective area scalability of the transducer are demonstrated by functional integration into an endoscope probe with a small radius of 3 mm and a large area (91.2×14 mm$^2$) non-invasive blood pressure sensor.

Ultrasound offers real-time imaging of deep-lying tissues, organs, and blood flow, in a safe and non-invasive way. It is the most widely used medical imaging modality in terms of number of images created annually[1–3]. Where current ultrasound systems require pointing and positioning by a sonographer, patches of flexible and large-sized ultrasound arrays enable hands-free imaging and offer a solution for short and long-term monitoring applications. With some notable exceptions[4,5], most prototypes of ultrasound patches are typically made by micro-assembly of individual piezoelectric transducer materials onto a flexible or stretchable substrate[6–10]. The rigid islands contain the functional ultrasound transducers, while thin electrodes in-between provide mechanical flexibility allowing the patch to conform to nonplanar surfaces. Spring-like metal interconnect lines and/or liquid metals can be used to provide stretchability. This approach suffers from a number of fundamental trade-offs. Firstly, typically less than 50% of the total area contains functional ultrasound transducers. This compromises the achievable image quality. Secondly, the use of PZT and PZT-based composites requires backing layers that are currently not included – this results in a relatively low bandwidth and thus axial (i.e., depth) resolution. Thirdly, the component assembly fabrication technique makes scaling the arrays to larger sizes and higher densities prohibitively expensive. Whereas conventional ultrasound transducers nowadays consist of 100 to 10,000 elements, the number of elements in flexible and wearable ultrasound transducers published

[1]Acoustics & Underwater Warfare, TNO, The Hague, The Netherlands. [2]Holst Centre, TNO, High Tech Campus 31, 5656 AE Eindhoven, The Netherlands. [3]Human Performance, TNO, Soesterberg, The Netherlands. [4]Medical Ultrasound Imaging Center, Department of Medical Imaging, Radboud university medical centre, Nijmegen, The Netherlands. [5]Physics of Fluids Group, Techmed Centre, Twente University, Enschede, the Netherlands. [6]These authors contributed equally: Paul L. M. J. van Neer, Laurens C. J. M. Peters. ✉e-mail: gerwin.gelinck@tno.nl

to date ranges from 10 to 256. Finally, the use of lead in these ultrasound transducers is to be avoided. We have developed an inherently flexible ultrasound transducer technology based on thin films of the biocompatible, lead-free piezoelectric polymer, P(VDF-TrFE)[11,12], to circumvent these shortcomings. By using thermal embossing in combination with foil lamination, we create densely packed (>64%) pillar structures that are 40 μm wide and ~80 μm high. Each pillar can be individually addressed or grouped, depending on the electrode layout. The pillar structure brings about two advantages. (1) It mechanically isolates neighboring elements. This strongly reduces acoustical crosstalk between neighboring acoustic elements, which is highly beneficial from a performance perspective[13]. (2) It increases the mechanical flexibility of the overall array[14] (see Supplementary Information (SI) Note 1 on mechanical considerations). Lateral dimensions of the P(VDF-TrFE) pillars were chosen to avoid Lamb waves. By varying the height of the P(VDF-TrFE) pillars, we can tune the operating frequency. In this work, we focus on the 5–10 MHz range, and specifically target the carotid artery[15], but note that this frequency range is also used in a wide range of conventional (e.g., for the imaging of organs in the abdomen, the neck and breast, and of children)[16] and endoscopic ultrasound applications (transesophageal, transrectal and transvaginal imaging)[2]. The resulting total thickness of our PillarWave™ ultrasound transducer is only 100 μm. With a value of 3.7 MRayl, the measured acoustic impedance of the P(VDF-TrFE) pillars is close to that of human tissue. Hence, the transmission coefficient of a bare piece of regular piezomaterial (35 MRayl) into tissue is 0.082, whereas for bare P(VDF-TrFE) on tissue, the transmission coefficient is 0.58. Therefore, the bandwidths typical for medical imaging can be reached without the need for matching layers or backings and the ringing of the transducer will be very low. Integrated into a linear array containing $2 \times 64$ elements (element size 175 μm × 2.5 mm) with a transmit aperture surface area of $11.5 \times 2.5$ mm², pulse-echo efficiencies of 0.2 are achieved—on par with competing approaches—and peak pressures just above 1 MPa are achieved (see Supplementary Table 1 and Note 2 for benchmarking). The imaging performance of the linear array is evaluated using a commercial tissue-mimicking ultrasound phantom. High-quality ultrasound images are obtained, from which important parameters such as the axial and lateral resolutions are extracted. The mechanical flexibility is shown by simple pulse-echo measurements of an array wrapped around a 6-mm endoscope. Finally, to demonstrate the large-area array fabrication possibilities of our manufacturing technology, an array transducer is fabricated with an aperture size of $9.1 \times 1.4$ cm² aimed at non-invasive ultrasonic blood pressure sensing on the carotid artery. Its performance is shown on a carotid phantom and the in vivo performance on a volunteer.

## Results

### Fabrication of P(VDF-TrFE) transducers on thin and flexible plastic substrates

In this work, we have used commercially available P(VDF-TrFE) as the piezoelectric material for the transducer. P(VDF), its copolymers (including P(VDF-TrFE)), and composites are well-known for their strong electroactive responses (piezoelectric, ferroelectric, pyroelectric), high dielectric breakdown strength and high dielectric constant[11]. This has resulted in numerous device prototypes, including hydrophones[17], energy harvesting devices[18,19], proximity sensors[20], ferroelectric memories[21], and actuators[22–25]. Recently, Qualcomm commercialized under-display fingerprint sensors using ultrasonic detection with P(VDF-TrFE)[26]. In this work, we specifically use the 80/20 TrFE copolymer of P(VDF) as this ratio gives the largest piezoelectric response for these materials.

A *ca.* 40-μm-thick P(VDF-TrFE) film was laminated on a polyimide substrate (thickness 14 μm) containing thin bottom electrodes. This P(VDF-TrFE) film is structured through hot embossing with a PDMS stamp, resulting in highly uniform pillars with a height of 70 μm on top

of a residual P(VDF-TrFE) film of about 10 μm (Fig. 1a). The embossing step was followed by laminating a second P(VDF-TrFE) film of about 10 μm on top, providing a flat surface for the deposition of a patterned top electrode (Fig. 1b).

A cross section of the transducer is shown in Fig. 1c. Both corona and contact poling were used to electrically polarize the structured piezoelectric layer. Corona poling gave less electrical breakdown and higher $d_{33}$ values compared to contact poling. To promote uniformity over large areas, we employ a corona procedure where the sample constantly moves back and forward under a set of corona wires. The device fabrication is finished by the physical vapor deposition of a patterned common top electrode. Optionally, a parylene C encapsulation layer was used. All steps are performed on substrate sizes up to $15 \times 15$ cm² or larger ($32 \times 35$ cm²) using glass as a temporary carrier.

The transducer properties strongly depend on the geometry of the structured piezoelectric, and can be tuned by both the thickness of the initially laminated P(VDF-TrFE) and the stamp design. The piezoelectric thickness (~80 μm), and the thicknesses of the substrate and encapsulation determine the resonance frequency. The kerf, i.e., the distance between the pillars, is ideally minimized through the stamp design, as it increases the active area and, therefore, the transmit and receive efficiency of the transducer area. We have found that a kerf smaller than 10 μm compromises pattern reproducibility over larger areas as a result of the so-called wall collapse of the PDMS stamp[27], see also SI Note 3. We therefore investigated the pillar shape and found that rectangular structures are more sensitive to wall collapse than hexagons, due to the decreased rib-length of the hexagonal pattern. Because of this, we have used hexagonal pillar arrays (40 μm diameter, 10 μm kerf) throughout this work, with a resulting active area of >64%. Further reduction of the kerf should be possible but would require additional process optimization.

The piezoelectric response is electrically measured using a Berlincourt setup, with which we consistently measured $d_{33}$ values of 25–29 pC/N, on par with state-of-the-art P(VDF-TrFE) piezoelectric performance. This indicates that the embossing, lamination, and heating steps do not significantly degrade the piezoelectric response of the polymer material. The measured compressional wave speed of 2100 m/s was slightly lower than the typically reported value of 2400 m/s for P(VDF-TrFE)[24]. This leads to a better acoustic impedance match with tissue, resulting in a higher transmission coefficient and increased bandwidth but also in a slightly lower piezoelectric coupling factor. Details of the fabrication are provided in the Methods section.

With a total thickness of around 100 μm, the transducer is inherently bendable due to the pillar structure of the array and the absence of ceramic layers (Fig. 1d). It can be bent to radii far below 1 mm. Figure 1e shows a transducer that is wrapped around a balloon catheter aimed at intravascular use. In Fig. 1f, an array transducer is directly attached to the human body, i.e., the neck, for carotid blood pressure measurements (see below for more information). The mechanical flexibility and low weight also makes it comfortable to wear.

### Characteristics of single-element transducers

As a first step, we characterized large circular single elements. By patterning the top and bottom electrodes in the form of circles with a diameter of 12 mm, we capture the collective response of over 52,000 pillars. Figure 2a shows the transmit efficiency, measured by scanning the produced acoustic field in water using a needle hydrophone and applying inverse wavefield extrapolation to obtain the produced acoustic field at the surface of the transducer[28], as a function of frequency. Figure 2b shows the frequency response of the receive sensitivity as determined from the transmit transfer function using reciprocity theory and electrical impedance measurements[29]. The transmit transfer function showed a peak of 1.4 kPa at 8.9 MHz, whereas the receive transfer function showed a maximum of 67 μV/Pa.

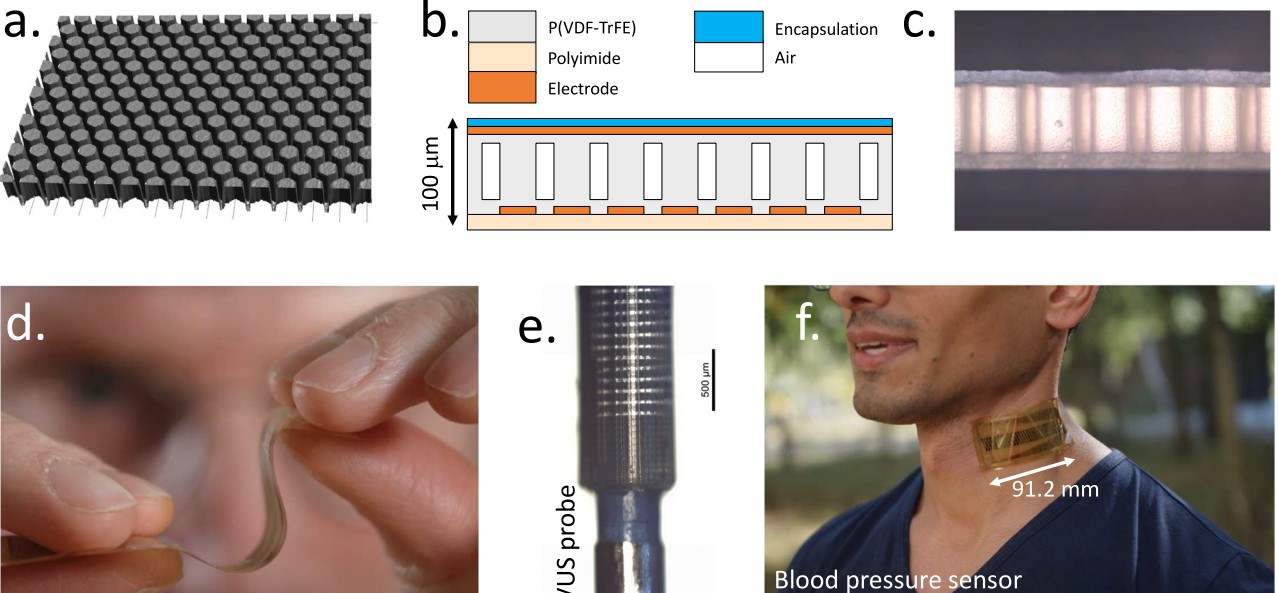

**Fig. 1 | PillarWave™ technology. a** Confocal microscope image of the P(VDF-TrFE) film directly after embossing. **b** Schematic cross section of the flexible ultrasound transducers. On top of a polyimide foil (thickness 14 µm) and a patterned molybdenum-aluminum electrode (500 nm thick) a ca. 40 µm P(VDF-TrFE) film is embossed resulting in 3D structures that are ca. 70 µm high with a residual layer of ca. 10 µm below. On top of the 3D structures, a 10 µm P(VDF-TrFE) film is laminated at elevated temperatures, whereafter a 500 nm molybdenum-aluminum top-electrode is deposited. The stack is finished with an isolating and flexible encapsulation film. **c** Side view of the complete transducer after lamination of the top electrode. The white areas are the P(VDF-TrFE) film. The orangish areas are air-filled gaps between the pillars. **d** Photograph of the finished ultrasound transducer foil, illustrating its thinness of 0.1 mm and mechanical flexibility. **e** Transducer foil wrapped around the inner wire of a dilatation catheter for percutaneous transluminal angioplasty (PTCA) (Blue Medical Force NC) that has a radius of 0.25 mm for intravascular ultrasound. **f** Ultrasound transducer foil placed in the neck on top of the carotid artery for blood pressure measurements.

The −6 dB frequency bandwidths, were measured to be 65 and 64% for the transmit and receive transfer functions, respectively.

By scanning the pressure generated by applying voltage pulses with different magnitudes along the x-axis (lateral axis) and the y-axis (elevation axis), we obtain a good impression of the surface uniformity as well as the linearity of the response. Figure 2c shows the peak transmit efficiency measured at 8.9 MHz as a function of the location at the transducer surface. The surface averaged peak transmit transfer was 1.4 kPa/V. The peak-to-peak efficiency variation over the total element area of 113 mm² was within ±12%, except for two local spots with lower efficiencies. These spots result from small air bubbles on the surface of the transducer during the actual characterization measurement. Long-range random variations comes from process imperfections, yielding a spread in pillar geometry and/or poling efficiency. The circular ripple pattern in Fig. 2c is an artifact of the wavefield extrapolation algorithm that was used in combination with the finite transducer area[28].

## Modeling of transducer elements

To predict the acoustic device performance, modeling was performed using a modified KLM model[30]. A number of properties and input parameters were directly measured or taken from literature: the density of P(VDF-TrFE), the film thickness, the compressional wave speed, and the active diameter. Using these parameters together with reported values of the coupling factor, the dielectric permittivity, the dielectric loss, and the mechanical loss, the model did not accurately describe the electrical and acoustical behavior of our transducers. We found that the large and frequency-dependent losses of P(VDF-TrFE) should be taken into account, in line with the results of refs. 31,32. For more details of the simulations as well as the experimental input parameters, including the derivation of their frequency response, we refer to Supplementary Information Note 3.

Figure 2a shows the modeled and measured transmit transfer functions in water. The modeled and experimental resonance frequencies were 8.6 and 8.8 MHz respectively, whereas the modeled and experimental mean peak transmit transfers were 1.5 and 1.4 kPa/V. The modeled and experimental frequency bandwidths at the −6 dB level were 58 and 65%, respectively. A good match was obtained between the modeled and measured transfer functions. The measured transfer function averaged over the active surface area was, on average, 15% lower than the modeled transfer function. The obtained coupling factor $(k_t)$ was 0.185, the relative permittivity $(K^S_{33})$ was 6.5−0.0088*frequency (MHz), the dielectric loss tangent $(tan(\delta_e))$ was 0.075 + 0.04*frequency (MHz), and the mechanical loss tangent $(tan(\delta_m))$ was found to be 0.125. The pulse-echo insertion loss was calculated to be -20.4 dB. For more information, see Supplementary Information Note 3. Figures 2d, e show the modeled and measured magnitude and phase of the electrical impedance as a function of frequency for the condition where the transducer was in air or the front face was in contact with oil. Here, electrically isolating oil was used instead of water, since the electrical isolation of this early prototype was suboptimal. The acoustic impedance of the oil was similar to that of water. The shape of the electrical impedance curves are dominated by the high amplitude transmission coefficient (0.58) from P(VDF-TrFE) to water/oil, due to the low acoustic impedance of P(VDF-TrFE) (3.7 MRayl) relative to the acoustic impedance of water/oil (1.5 MRayl) and therefore center frequency and transfer function of the measurements in oil are considered representative. The higher acoustic attenuation in the oil does add some damping, but it is a very minor effect. Part of the effective properties of the P(VDF-TrFE) based PillarWave™ transducer was determined based on the electrical impedance measurements in the air using the fitting procedure described in the Methods section (and Supplementary Information Note 4). In the case of oil, the match between model and experiments

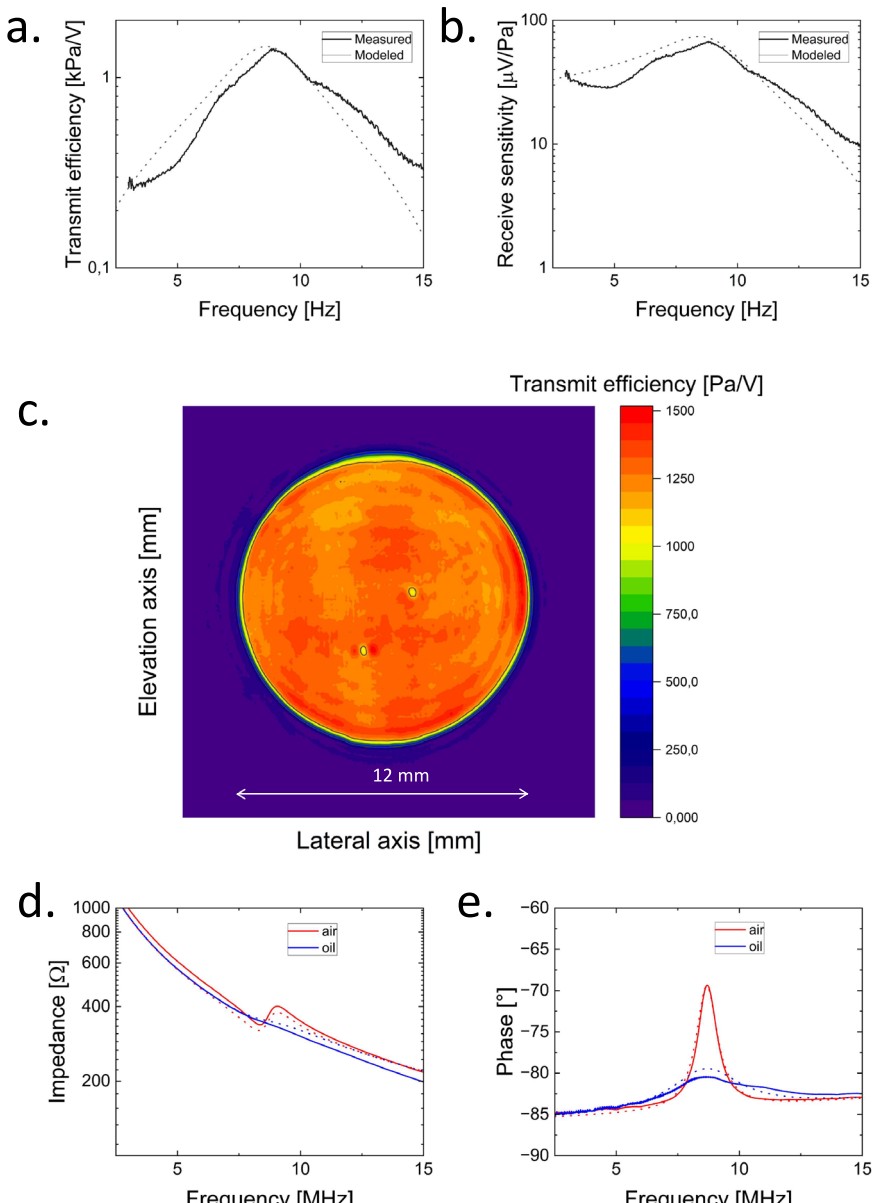

**Fig. 2 | Characteristics of a single element. a** Transmit efficiency and **b** receive sensitivity as a function of frequency in water. **c** Transmit efficiency at 8.93 MHz as a function of location at transducer surface measured in water. **d** Magnitude and **e** phase of the electrical impedance as a function of frequency for the condition in which the active surface area of the transducer is in contact with air or oil. Experimental results are indicated by solid lines. Modeled results use dotted lines.

was also good. Regarding the phase of the electrical impedance, the difference between the model and experimental results was <1°. With respect to the magnitude of the electrical impedance, the difference between model and experimental results is <16%. This validated model was used to design the other transducers described next.

### Pulse-echo measurements while bent around a 6-mm endoscopic probe

Linear array transducers optimized for the ultrasonic imaging of the carotid artery were fabricated using the manufacturing process described above. The array consists of separate parallel transmit and receive apertures with 64 elements each. Each aperture had a size of $11.5 \times 2.5$ mm², the element pitch was 180 μm. Its performance was measured in water using a hydrophone setup. Figure 3a shows a picture of the slightly bent flexible array. Figure 3b shows a photograph of the 128-channel array integrated on a 6-mm ultrasound probe for endoscopic ultrasound (EUS). Figure 3c shows pulse-echo

measurement of the array while bent around the 6-mm EUS probe. The signal at 25 μs is an echo of a 2.5 mm diameter metal cylinder, whereas the response at 35 μs originates from the edges of the water tank. Low voltages (<20 $V_{pp}$) were used. These measurements illustrate that the array continues to function in pulse-echo mode while strongly bent. Preliminary experiments show an effect of the curvature on the wavefield, and a design of an optimized EUS device will need to take into account this effect.

### High-resolution imaging with the 128-element linear array transducer

Figure 4a shows the measured average transmit and receive transfer functions as a function of frequency using the 128-element linear array transducer described above and shown in Fig. 3a. In transmission, the center frequency of the array transducer was 8.2 MHz, and the frequency bandwidth was measured to be 78% at the −6 dB level. The peak transmit transfer was 1.3 kPa/V. In reception, a peak receives a transfer

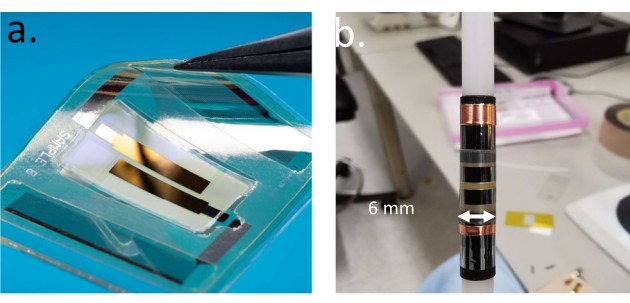

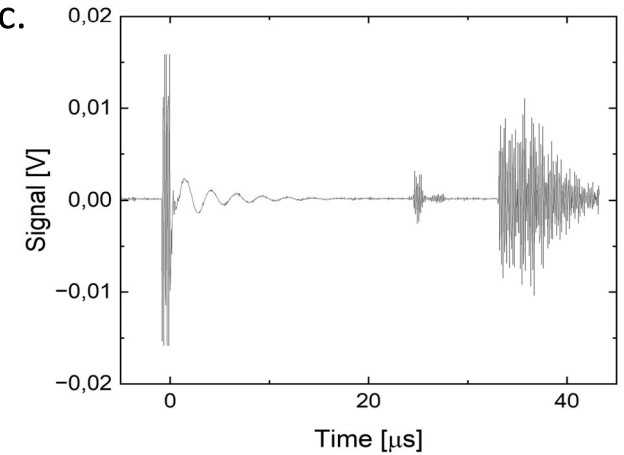

**Fig. 3 | Characteristics of 128-element flexible polymer array transducer.**
**a** Photograph of the flexible array while slightly bent. The design consists of two 64-element arrays—their locations are indicated by the gold-colored top electrodes –, one used in transmission, one used in reception. **b** Photograph of the array integrated on a 6-mm EUS probe. **c** Pulse-echo signal of the array wrapped around the EUS probe measured in water.

of 150 µV/Pa at 7.2 MHz and a −6 dB bandwidth of >107% were obtained. These numbers are even a little higher than the results of the single transducer presented in Fig. 2 due to slight process optimizations. Taking the maxima of the transmit and receive efficiency of 1.3 kPa/V and 150 µV/Pa, we obtain a maximum pulse-echo efficiency of 0.2. The peak transmit transfer measured at 8.2 MHz as a function of location at the transducer surface is shown in Fig. 4b. The per-element performance variation (peak-to-peak) was ±10%. The peak transmit transfer was improved from 1.3 kPa/V to 5.2 kPa/V by the application of electrical tuning. Peak pressures in excess of 1.05 MPa were measured at a focal distance of 4 cm with an excitation voltage of 95 V. The noise equivalent pressure of the array, when connected to a Verasonics Vantage machine, was 1.6 Pa over a 3–10 MHz bandwidth. Even after prolonged periods of time at these high voltages, no device degradation was observed. The element-to-element crosstalk was ~-31 dB. Its instantaneous nature implied it was mainly electrical, indicating acoustic crosstalk is effectively reduced (see SI Note 5). The transmitter-to-receiver-array crosstalk was −69 to −63 dB, depending on the frequency.

Figure 4c shows a typical ultrasonic B-mode image obtained using a tissue-mimicking phantom (040GSE, CIRS, Norfolk, Virginia, USA). Such phantoms provide an invaluable approach to objective, quantitative evaluation of image quality characteristics. The image was obtained using the 128-linear array on so-called plane wave compounding (of nine plane wave transmissions)[33] in combination with delay-and-sum beamforming in reception[34]. The reflections of the nylon wires are clearly visible with smooth and sharp point-spread functions (PSFs). The imaging capabilities of PillarWave in Fig. 4c were quantified using international standardized methods IEC 61391.1:2006 and IEC 61391-2:2006. For the nylon wire at [−8, 20] mm, the lateral

width of the PSF was 0.63 and 1.3 mm at the −6 and −20 dB levels, respectively. The corresponding axial lengths (along the depth axis) of the PSF were 0.23 and 0.92 mm at the −6 and −20 dB levels, respectively. The echoes of the wires at [20, 30] mm that are typically used to determine the axial/lateral resolution are also clearly visible and well separated. The hyperechoic region at [−20, 30] mm is clearly visible, as is the 10 kPa elasticity target at [5, 15] mm. The image was obtained with a frame rate of 15 frames per second (fps)—no averaging was applied. The frame rate was limited by the real-time processing implemented in the Verasonics Vantage machine. B-mode imaging was performed with frame rates up to 4 kHz with two angles used for the plane wave compounding. These results demonstrate the capability of the fabricated flexible array to produce high-quality real-time medical ultrasound B-mode images. Supplementary Data 1 shows a comparison between our flexible array and previously reported work. The comparison is based on a large number of geometrical (thickness, number of transducer elements, their pitch and density, etc.) and performance parameters (bandwidth, penetration depth, image resolution, etc.) as ultrasound arrays are notoriously difficult to compare.

**Large aperture array transducer (91.2 × 14 mm²) for blood pressure monitoring applications**
To demonstrate the scalability to a large area of our technology, an array with an exceptionally large active aperture of 91.2 × 14 mm² was realized (see Fig. 5a). The array consisted of four staggered rows of 32 transducer elements. Each transducer element had a size of 1.6 × 3.2 mm² and comprised approximately 2365 pillars. The element pitch was 2.8 and 3.6 mm in the lateral and elevation directions, respectively. The design of the array was optimized to make sure at least one combination of transmitter and receiver elements would be located optimally with respect to the carotid artery (i.e., the acoustic beam will cross-sect the center of the carotid), independent of the patient's head movement. The fabrication process steps are identical to the imaging array discussed earlier, only a different electrode design is required. This illustrates the versatility of our technology. As shown in Fig. 1f, the ca. 10 cm long array conforms well to the shape of the human neck. This would not be possible with a rigid ultrasound transducer. It would suffer from poor acoustic coupling because of its rigid surfaces not being able to accommodate the curvilinear shape of a human neck without the use of excessive pressure.

Rows 1 and 3 were simultaneously excited in transmission, and row 2 was read-out in parallel. The center frequency of the device was 8.2 MHz. The measurements shown in Fig. 5b indicate that 95% of the elements were working, with a 50% variation in peak transfer functions (transmit: 0.6–1.3 kPa/V, receive 50–100 µV/V). The array was tested on a home-built in vitro carotid phantom (see Methods for more details)[35]. The carotid vessel wall was tracked by cross-correlating the echoes of the anterior and posterior walls over time. The resulting vessel diameters were subsequently converted into blood pressure waveforms using a calibration with a blood pressure sensor, see Fig. 5c, using an established conversion method[6]. A high correspondence between the measured and reference pressures was observed: the difference is less than 5%. Figure 5d shows in vivo data of the carotid artery. Clear echoes of the vessel walls are observed. The temporal variation, or pulsation, of the proximal and distal wall echoes resulting from the heart beating, can clearly be discerned.

## Discussion
We reported a flexible transducer technology for wearable ultrasound applications applicable to large-area. The technology is based on microstructured thin films of P(VDF-TrFE) with a thickness of only ca. 80 micrometers. The pillar structure leads to low acoustical crosstalk. It furthermore increases the mechanical flexibility compared to an unstructured P(VDF-TrFE) film. The low acoustic impedance of P(VDF-TrFE) enables a large frequency bandwidth and a high

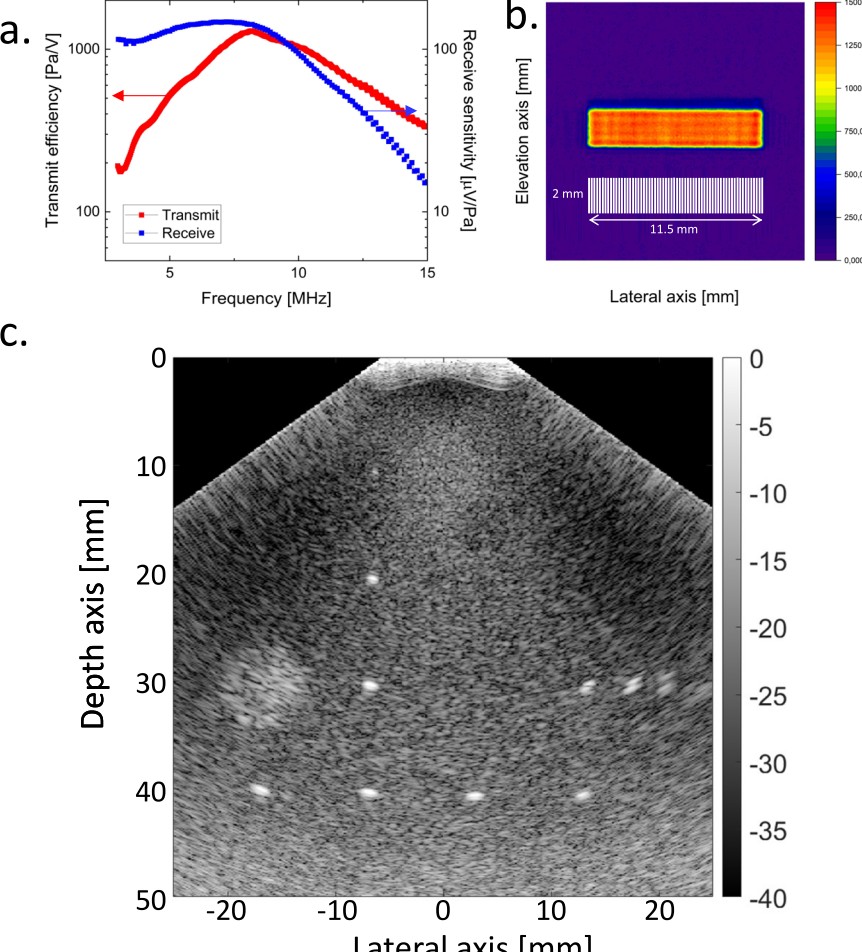

**Fig. 4 | High-resolution imaging of tissue-mimicking phantom using a 128-element flexible polymer array transducer. a** Measured transmit and receive transfer functions versus frequency. **b** Area uniformity of the peak transmit transfer at 8.2 MHz at the transducer surface. The color scale indicates the peak transmit transfer in Pa/V. **c** B-mode image captured with plane wave compounding. The gray scale indicates the intensity in dB.

axial resolution without the necessity of matching or backing layers. The potential of the technology is demonstrated in three applications, each showing a specific unique advantage of the PillarWave™ technology—an endoscopic ultrasound (EUS) transducer that remains functional while mechanically curved over a small radius of 3 mm, b-mode imaging of the carotid showing good spatial resolution, and a blood pressure sensor that illustrates the potential to scale to a large area. We point out that in all cases, the same fabrication process was used, using a single P(VDF-TrFE) pillar geometry. The size of the ultrasound elements is actually determined by the design of the top and bottom electrodes. This permits the simultaneous fabrication of multiple transducers with different designs on a single substrate – as long as these use equal pillar heights (i.e., operating frequencies). Our transducer technology is expected to scale over the full range of medically relevant ultrasound frequencies (1–60 MHz). Moreover, the manufacturing technology is based on a large-area fabrication process and avoids complicated assembly technology. Currently, the maximum substrate size is $32 \times 35 \, cm^2$; however, it is expected that—similar to display production—substrate sizes of a few square meters may be used in the future. This could dramatically lower production costs and allow for a rapid, cost-effective route to mass-manufactured flexible large-area ultrasound arrays. Currently, we are in the process of integrating the polymer transducer technology reported here with a thin-film transistor backplane, with the aim to reduce the number of interconnections and, thus, the cost of the addressing electronics.

Our flexible large-area transducer technology is aimed at wearable ultrasound applications. Initially, these could be mainly inside clinics. However, applications of medical ultrasound outside of clinics are emerging, such as preventive examinations at the general practitioner, monitoring during (extreme) exercise, or monitoring in the home environment (e.g., of pregnant women). The combination of high performance, low cost, scalability, flexibility, and lead-free components makes this technology uniquely suited for these new applications.

## Methods

### Stamp fabrication and embossing process

The stamp master is created on a separate glass substrate by lithographically structuring SU-8 2050 with pillar structures with the desired shape (e.g., square, hexagon, and circle), pattern, and height. An inverse soft stamp of the pillar structure is made using PDMS (Sylgard 184, Dow Corning, Michigan, USA). The use of a soft stamp is generally favored over a hard stamp for large-area embossing[36]. Using a soft (PDMS) stamp, it was possible to gently release the stamp, starting from the edge, and slowly working our way across the panel. When a hard (SU-8) stamp was used, release became problematic, even for small substrate sizes.

### Transducer fabrication

The PillarWave™ transducer is fabricated through a series of patterned electrode deposition, lamination, and embossing steps. On a

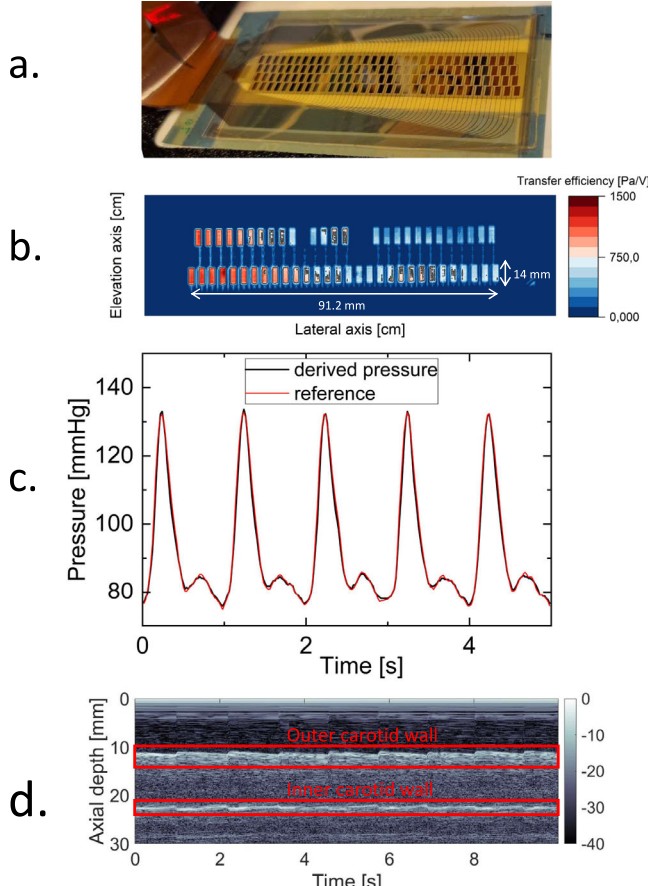

**Fig. 5 | Large-area flexible ultrasonic blood pressure sensor. a** Photograph of a large area of flexible ultrasonic blood pressure sensor while still on the support glass. See Fig. 1f for a photograph of the blood sensor placed in the neck. **b** Transmit transfer in Pa/V at the resonance frequency of 8.2 MHz of transmit elements, obtained using hydrophone measurements. **c** The pressure waveforms derived from the measured carotid phantom vessel diameters as a function of time. The diastolic pressure $p_d$ was taken from the reference blood pressure sensor, and the vessel stiffness $\alpha$ was fitted such that the obtained systolic peak pressure matched with the reference blood pressure. The red curve shows the blood pressure measured using a reference blood pressure meter. **d** Recorded in vivo ultrasound data of the carotid of a healthy volunteer of the optimally positioned array element.

temporary glass carrier with a 14 μm thick spin-on polyimide film (PI 2611, HD Microsystems, Neu-Isenburg, Germany), a bottom electrode of 500 nm MoCr-Al-MoCr is sputtered and structured lithographically. Subsequently, a 40 μm sheet of P(VDF-TrFE) (80/20 mol, PolyK, Philipsburg, USA) is laminated on the polyimide/electrode substrate. A short CF4/O2 plasma treatment directly prior to lamination was found to increase the adhesion strength to 200 mN/mm, which is sufficiently high to prevent delamination: this treatment provided long-lasting mechanical stability in all transducers studied. Next, the PMDS stamp is pressed into the P(VDF-TrFE) film that is heated to 160 °C, just above the melting temperature of the P(VDF-TrFE). After releasing the PDMS stamp, a second P(VDF-TrFE) film is laminated on top of the arrays of P(VDF-TrFE) pillars. This second film is softened by heating so it partly squeezes in between the pillars on the bottom substrate, provides us with a flat surface for the subsequent deposition of the electrode. Thereafter, the stack is annealed for 1 h at 140 °C[37]. The piezoelectric layer is polarized using corona poling (custom-built setup, 15 kV wires at 2 cm, no grid), whereafter a common top electrode (MoCr-Al-MoCr) is sputtered using a shadow mask. Finally, the transducer is mechanically de-bonded from the glass. A photograph of a 15 × 15 cm glass plate containing several arrays is shown in the SI Note 6.

## Characterization of discrete transducers

The piezoelectric $d_{33}$ coefficient is measured using a Berlincourt setup (d33 PiezoMeter System, Piezotest, London, United Kingdom). The electrical impedance of each transducer was measured using a vector impedance meter (ZVRE, Rhode & Schwarz, Munich, Germany). The acoustic wavefields produced by the flexible single element or array transducers were measured using a hydrophone (diameter 0.2 mm, Precision Acoustics, Dorchester, UK) mounted in an A3200 Npaq, Aerotech Inc. system (Pittsburg PA, USA). The transducer was excited by linear chirp signals of various amplitudes, lengths and frequencies (−6 dB bandwidth 3–12 MHz) generated by an arbitrary waveform generator (33621 A, Agilent Technologies, Loveland, Colorado, USA) and amplified by a power amplifier (75A250A, AR RF/Microwave Instrumentation, Southerton, PA, USA), or a programmable ultrasound system (Vantage 128, Verasonics, Kirkland, USA). The signals received by the hydrophone were amplified by an amplifier (5900, Olympus NDT Inc., Waltham, MA, USA) and digitized (MI.4032, Spectrum Instrumentation, Grosshansdorf, Germany). The excitation voltage over the electrodes was read-out using an electrical probe and digitized.

The receive transfer functions and angular sensitivities of the flexible single element or array transducers were measured using a custom-calibrated source transducer (V311, Olympus NDT Inc., Waltham, MA, USA). The source transducer was excited by signals (linear chirps, −6 dB bandwidth 3–12 MHz, various amplitudes and lengths) generated by the arbitrary waveform generator. The pressure signals were received by the flexible single-element or array transducers, amplified by custom-designed trans-impedance amplifiers and further amplified and digitized by the programmable ultrasound system. The custom-calibrated source transducer was calibrated using a pulse-echo method[38,39].

Although no endurance testing was performed, the prototypes operated in the lab for more than a year without performance degradation.

## Measurement and characterization of material properties

The thickness of the P(VDF-TrFE) was measured using profilometry (Bruker Dektak XT, Massachusetts, USA). The compressional wave speed was measured independently using an acoustic transmission measurement. Here, a high-frequency wave was transmitted (V113-RM, Olympus NDT Inc., Waltham, MA, USA) driven by an arbitrary waveform generator (33250, Agilent Technologies, Loveland, Colorado, USA), sending a 6 MHz burst sinusoidal wave along the thickness direction of the array. The effective compressional wave speed was calculated using the thickness and the arrival time measured using an oscilloscope (DSO6032A, Agilent Technologies, Loveland, Colorado, USA) of said wave. The active radius of the transducer was determined by measuring the acoustic field in a plane perpendicular to the transducer axis, and backpropagating said field to the transducer surface. The density was provided by the manufacturer.

## Tissue-mimicking phantom used in high-resolution imaging experiments

The high-resolution images (Fig. 4) were obtained using a commercial tissue-mimicking phantom (040GSE, CIRS, Norfolk, Virginia, USA). This phantom has been designed to optimally mimic the acoustic properties of tissue: its sound speed of 1540 m/s and sound attenuation of 0.7 dB/(MHz.cm) correspond well with average human tissue properties. The phantom contains nylon filament wire targets, to simulate small but strong reflectors in the human body. Furthermore, it has hyperechoic targets, optimized to provide an echo with a predefined relative strength compared to the speckle background and elasticity targets, areas with predefined elasticity values. All features of the phantom are clearly visible in the recorded ultrasound image, and used to quantify the performance of the array.

## Image readout and processing for 128-element linear ultrasound array

To perform B-mode imaging with the flexible array transducers a programmable ultrasound system (Verasonics Vantage) was connected to the flexible transducers. The ultrasound system generated the excitation signals. In reception, the signals were first amplified by custom-designed trans-impedance amplifiers before being routed to the programmable ultrasound system for further amplification and digitization. The post-processing of the recorded radio-frequency signals consisted of the following steps:

1. Subtraction of the average signal level
2. Chirp compression
3. Time windowing
4. Time-frequency filtering
5. Application of imaging algorithm, either:
a. Wavenumber-frequency domain mapping (Stolt migration)[28,40]
b. Plane wave compounding[33]
6. Wavenumber-frequency filtering[40]

More details can be found in SI Note 7.

## Carotid phantom used in blood pressure measurements

To evaluate the prototype array for blood pressure measurements a home-built in vitro carotid phantom was used. A PVA solution (all % by weight) of polyvinyl alcohol (PVA) Cyrogel (10%), distilled water (40%), ethylene glycol (10%), and silica gel particles (2%) was heated to 85 °C until a homogeneous liquid was formed. Next, the solution was poured into a mold with a circular inner lumen (10 mm) with a centrally placed rod (5 mm) to form a carotid vessel with an outer diameter of 10 mm and a lumen diameter of 5 mm. The mold was subjected to three cycles of freezing (−25 °C) and thawing (21 °C), each 16 and 8 h, respectively, to solidify the vessel. During the last freeze-thaw cycle, the vessel phantom was immersed in another PVA solution to create surrounding tissue and minimize lumen translational motion. The echogenicity and the outer and luminal diameters of the carotid phantom are in the same range as human carotid arteries, while the elastic modulus of the PVA solution after three freeze-thaw cycles result in similar expansion levels of the carotid phantom as present under human physical conditions[35].

## In vivo carotid artery experiments

To evaluate the prototype array for blood pressure measurements, in vivo experiments were conducted with the approval of TNO's ethical review committee (IRB 2023-058). The aims of the study are: (1) Investigate the performance of the large-area flexible array on a human volunteer (in vivo), (2) Extract the carotid wall displacement from the ultrasound data, (3) Compare the data quality between earlier obtained in vitro data and the in vivo datasets. A sample size of $n = 1$ was chosen prior to the actual study. Inclusion criteria were defined upfront. The human subject was recruited randomly by posting flyers. No specific preparation was required of the human subject. The human subject was told the experimental process, and was not involved in data processing. Blinding to the investigators was not required since the sample size was one. The data were captured from a live subject at rest (healthy, male, age group 40–45 y, informed consent). These results are not specific to one sex or gender, and sex and gender aspects were not considered in the design study.

Data collection of the ultrasound signal was carried out with a Verasonics Vantage platform. The prototype array was temporarily placed in the neck of the subject, using a commercial ultrasound gel. The prototype array had four rows of piezoelements and a center frequency of 8.2 MHz. Rows 1 and 3 were simultaneously excited in transmission, and row 2 was read-out in parallel. The excitation consisted of 20-cycle chirps with a maximum amplitude of 10 V and a bandwidth of 3–12 MHz. The pulse repetition frequency was 50 Hz.

The ultrasound data was visualized in real-time using a simple imaging procedure consisting of a color plot with the envelope (obtained by taking the absolute of the Hilbert transform) of the recorded radio-frequency data of the 32 channels read-out in reception. The carotid vessel was manually located in the ultrasound data, by observing the rhythmic expansion and compression of the carotid wall echoes. This allowed us to select the channel with the optimal acoustic path through the carotid (through the center), the results of which are shown in Fig. 5d. More details can be found in SI Note 8.

**Statistics and reproducibility.** All attempts at data and sample replication were successful. During a measurement session of ca. 3 h, the in vivo carotid ultrasound measurements were taken for more than three times. All testing results showed high similarity.

## Reporting summary

Further information on research design is available in the Nature Portfolio Reporting Summary linked to this article.

## Data availability

All data supporting the findings of this study are available within the article, its supplementary files or in the repository (github.com/horchensl/pillarwave). Any additional requests for information can be directed to, and will be fulfilled by, the corresponding authors. Source data are provided with this paper.

## Code availability

Codes used to post-process the ultrasound data within this paper are available from the corresponding author upon request.

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

## Author contributions

L.C.J.M.P., P.L.M.J.v.N., A.W.F.V., E.J.W.M-S., R.G.F.A.V., G.d.H, J.-L.P.J.v.d.S., and G.H.G. designed and fabricated the ultrasound devices on foil. L.C.J.M.P., T.S., and E.J.W.M.-S. performed the poling, Berlincourt measurements, and the sound speed measurements. P.L.M.J.v.N., L.H., L.F., T.S. E.J.W.M.-S., and A.W.F.V. performed the acoustic measurements on the single elements and the 128-channel linear array. Simulations were performed and interpreted by P.L.M.J.v.N., L.H., L.F., T.S., E.J.W.M.-S., K.G., and A.W.F.V. B.P. and L.C.J.M.P. measured the ultrasound pulses of the ultrasound array while wrapped around the EUS. B.P., E.J.W.M.-S., and T.S. designed and realized the receive electronics (PCB plus amplifiers). L.F., L.H., K.G., A.E.C.M.S., M.M., J.M.M., and C.L.d.K. performed the blood pressure measurements and interpreted the data. L.C.J.M.P., P.L.M.J.v.N., and G.H.G. planned the research and interpreted the data. L.C.J.M.P., P.L.M.J.v.N., and G.H.G. wrote the manuscript. All authors commented on it.

## Competing interests

A patent application has been filed by the authors (L.C.J.M.P., J.-L.P.J.v.d.S., R.G.F.V., P.L.M.J.v.N., and G.H.G.) under the number EP3869575A1. This patent describes the technology and device structure of the ultrasound transducers used in this work. PillarWave is a trademark of TNO. The remaining authors declare no competing interests.
