## [Peer Review File · Nature Communications]

REVIEWER COMMENTS

Reviewer #1 (Remarks to the Author):

The manuscript reports on the fabrication and test of ultrasound (US) transducer-on-foil technology based on biocompatible, lead-free piezoelectric polymer, P(VDF-TrFE). This work's main novelty relies on the scalable production of flexible, safe, skin-compliant large arrays of US transducers, which have been applied and validated on phantoms, into an endoscope probe and on human skin for blood pressure sensing.

Most of the works in this field rely on lead-containing piezoelectric transducers and this work, therefore, emerge as one of the first attempts to move to a more sustainable, non-toxic and environmentally friendly example of US transducers technology. Despite being thinner and operated at lower voltages, the performance of the US array aligns with the state of the art.

I enjoyed reading this manuscript, which can be of great interest to the community working on wearable technology and ultrasound applications.

I have only a few minor questions/comments.

- The fabrication employs a soft PDMS replica of SU8 stamps. As documented in the supplementary section, this can lead to a lack of fidelity, buckling of the pillars, curved walls, and non-uniform thicknesses. The author should explain their choice of a soft stamp instead of a more rigid one (e.g. based directly on SU8).
- The map of the transmit efficiency in Figure 2 shows non-uniformities. Can the author elaborate more on the origin of the differences between pillars? Are those due to the fabrication or poling procedures (or both)?
- The formation of gas bubbles has been mentioned. How can this be prevented? Is the printing done under vacuum?
- More information on the dependence of the cross-talk on the residual layer thickness and distance among pillars should be given.

Reviewer #2 (Remarks to the Author):

This manuscript presents a flexible ultrasonic device based on P(VDF-TrFE), featuring a microstructured thin film with a large aperture array. While certain distinctive aspects are highlighted by the authors—including the P(VDF-TrFE)-based transducer, pillar structure within each element, and the array's large

area/aperture, to name a few—the novelty of the study appears constrained. These emphasized points have been documented in previous studies, and, as such, do not project a substantial level of innovation within this work. In addition, the application of monitoring blood pressure waveforms has been proficiently demonstrated by previous wearable ultrasound devices. The manuscript would also benefit from the inclusion of additional experimental results to lend greater credence to the conclusions drawn. Detailed comments are as follows.

1. In Line 19, Page 2, the author mentioned that containing lead in the ultrasound transducer is a weakness. However, the transducer array is encapsulated well and PZT material would not touch to human tissue. People would not have chance to touch to lead. The reviewers are suggested to revise the content or provide other weakness of current ultrasound probe here. Similar to the sentence in Line 48, Page 3.
2. In Line 38, Page 3, the reference numbers of 6 to 10 have some typos. Please figure them out.
3. P(VDF-TrFE) is often used as a receiver due to its low impedance and wide bandwidth. However, its poor transmission performance (having a much lower electromechanical coupling coefficient compared to piezoelectric ceramics) generally disqualifies it as a transmitter. Why did the authors choose to use P(VDF-TrFE) for the flexible ultrasound probe used for sensing and imaging? How did the authors overcome the intrinsic disadvantages of P(VDF-TrFE)?
4. In Line 54, Page 3, the authors mentioned the advantages of pillar structure in this study. However, those are not novel. In previous research, 1-3 composite transducers fabricated based on piezoelectric ceramics (such as PZT, PMN-PT) possess this structure, and the related arrays also have very small cross-talk (for instance: -70 dB in Science Advances, 2018, 4(3): eaar3979). Additionally, researchers have increased mechanical flexibility by filling the kerf of PZT pillars with silicone elastomer (IEEE Transactions on Industrial Electronics, 2019, 67(8): 6955-6962). Therefore, the authors are recommended to introduce some new advantages?
5. Please add more explanation in Line 65, Page 4 about why it is possible to enhance the acoustic bandwidth without adding a backing layer for P(VDF-TrFE)? If no need for backing layer, how is the ringing effect being addressed?
6. In Line 95, Page 6, the authors mentioned corona poling was used for polarization. A potential issue with corona poling is the non-uniform polarization of the pillars. The uneven distribution of the corona charge could result in some sections of the material being polarized more strongly than others. This implies that a portion of the material might achieve a higher polarization intensity than other sections. Was this an issue in the study? Why not choose other methods that can achieve uniform polarization?
7. Please provide further explanation on how the thickness of P(VDF-TrFE), substrate, and encapsulation influences the resonance frequency? Additionally, factors influencing the resonance frequency of bulk PZT are not limited to thickness but also include the aspect ratio (thickness to width/diameter), as the aspect ratio can affect the vibration mode, thereby affecting the resonance frequency. Is this the case with P(VDF-TrFE)? Kindly elaborate.

8. In Fig.2, some crucial properties, such as electromechanical coupling coefficient, sensitivity (insertion loss), dielectric loss, and the SNR of a single element, need to be provided. Although the literature has related characterization results for P(VDF-TrFE), different fabrication methods can yield different performances, so actual characterization is necessary.

9. In Line 177 Page 9, the author mentioned the experiment was done in the oil because the acoustic impedance of the oil was similar to that of water. How about the acoustic attenuation of oil? If the acoustic attenuation of oil and water is different, does this influence the result of characterization? In addition, why can the experiments in Fig.3 be conducted in water, but not the ones in Fig. 2?

10. In Fig. 2d, why are the resonant and anti-resonant frequencies not distinct in oil (similar to Fig. 2e about the phase angle)? If the resonant frequency and piezoelectric properties would change under a load, then would the frequency and other properties also change when the device is tested on a person?

11. In Fig. 3c, because the element is soft, the acoustic field distribution emitted by the element could vary when the probe is placed on objects with different curvatures, resulting in different pulse-echoes intensities. Therefore, the authors are advised to perform more comprehensive characterizations demonstrating pulse-echo and SNR under different curvatures. The intensity of the reflected pulse-echo determines the range of curvature for practical applications.

12. The authors tested the pulse-echo while wrapped on an EUS probe in Fig.3, and characterized the spatial resolution of imaging in Fig.4, yet the final application (Fig.5) did not involve either the EUS probe or imaging, rendering the manuscript logically inconsistent. From reviewer's understanding, the characterization should serve the final application to some extent, to render the characterization meaningful.

13. A compelling demonstration/application should address a genuine, existing problem. The application in this manuscript, blood pressure waveform monitoring, has been achieved by previous conformal ultrasound devices. In addition, the test results in this study do not exhibit superior quality comparing with previous results. Moreover, the blood pressure waveform can be tested with just one element and does not require a large aperture array. Therefore, the authors are suggested to design another application and provide comprehensive data to showcase the unique advantages of a large aperture array that addresses a new clinical problem.

Response to the comments of the reviewers on the manuscript “Flexible large-area ultrasound arrays for medical applications using embossed polymer structures” by Van Neer *et al.* submitted to *Nature Communication*.

We thank the reviewers for their favorable comments and their thoughtful comments and suggestions to improve the manuscript. Below we repeat these comments and then provide our response and outline the changes made in the manuscript.

Reviewer #1:

The manuscript reports on the fabrication and test of ultrasound (US) transducer-on-foil technology based on biocompatible, lead-free piezoelectric polymer, P(VDF-TrFE). This work's main novelty relies on the scalable production of flexible, safe, skin-compliant large arrays of US transducers, which have been applied and validated on phantoms, into an endoscope probe and on human skin for blood pressure sensing.

Most of the works in this field rely on lead-containing piezoelectric transducers and this work, therefore, emerge as one of the first attempts to move to a more sustainable, non-toxic and environmentally friendly example of US transducers technology. Despite being thinner and operated at lower voltages, the performance of the US array aligns with the state of the art.

I enjoyed reading this manuscript, which can be of great interest to the community working on wearable technology and ultrasound applications.

I have only a few minor questions/comments.

Comment:

1. The fabrication employs a soft PDMS replica of SU8 stamps. As documented in the supplementary section, this can lead to a lack of fidelity, buckling of the pillars, curved walls, and non-uniform thicknesses. The author should explain their choice of a soft stamp instead of a more rigid one (e.g. based directly on SU8).

Reply:

The main reason to use a soft stamp is that a soft stamp can be released from the embossed structure on the glass substrate. Since we are creating a substantial amount of surface area by the fine pitch-high aspect ratio structures, the associated force to release the entire stamp simultaneously from the substrate becomes prohibitively high. We have in fact tried using the SU-8 master as the stamp itself, but the release became highly problematic, even for small aperture devices and after surface functionalization (silane-type) to enhance the release. With a soft stamp we are able to gently release the stamp, starting from edge, slowly working our way across the panel. This leads to highly reproducible results and reusability of the stamp.

A soft stamp has several other benefits. These are documented for instance by Zhou, W. in ‘Stamp Fabrication. Nanoimprint Lithography: An Enabling Process for Nanofabrication. Springer, Berlin, Heidelberg. https://doi.org/10.1007/978-3-642-34428-2_3) and by Viheriälä et al. (Nanoimprint Lithography - Next Generation Nanopatterning Methods for Nanophotonics Fabrication | IntechOpen). Stamp softness also allows imprinting of a large area in a single step, while maintaining a uniform residual layer. High uniformity is possible because a soft and flexible stamp conforms to the overall nonflatness of the substrate. When a particle is trapped between the stamp and the target substrate the soft stamp is soft and it will deform locally. This improves the yield of the process, since with a soft stamp a small particle destroys only a small area.

Implementation in manuscript:

We have included an additional comment in the Methods section in ‘Transducer fabrication’ to describe our results with the SU-8 stamp, and refer to Zhou’s book.

Comment:

2. The map of the transmit efficiency in Figure 2 shows non-uniformities. Can the author elaborate more on the origin of the differences between pillars? Are those due to the fabrication or poling procedures (or both)?

Reply:

We can distinguish different types of non-uniformities.

The first relates to the two bright yellow spots and originate from, as stated in the manuscript: “the presence of gas bubbles”. We realize now that this explanation is not very clear. The prototype transducer was mounted upside down under water. The prototype transducer transmitted pressure waves downwards towards the hydrophone. Although considerable effort was spent to prevent air entrapment on the surface of the transducer and hydrophone during the measurements (e.g. by removing gas bubbles using a soft cloth) a few air bubbles slipped through.

The long-range non-uniformity in peak transfer comes again from different sources.

Randomly shaped patterns are due to undesirable variation in pillar geometry and electrical poling efficiency. These effects are intertwined, and we therefore group them together as process imperfections, in the revised manuscript.

The circular ripple patterns are the result of how the data was obtained. The measurement consists of a hydrophone scan of the emitted acoustic wavefield over a finite area. The recorded pressure field dataset as a function of spatial coordinates x and y and time is then extrapolated back to the location of the transmitting transducer (the prototype). The wavefield extrapolation algorithm operates in the 3D frequency domain (temporal and spatial) (see G. F. Margrave, Numerical Methods of Exploration Seismology, University of Calgary, 2003). Due to the finite scan area in x and y coordinates in combination with the band limited signal that is emitted by the prototype transducer, the emitted pressure field at the surface of the source transducer can only be described by a finite (limited) number of spatial and temporal frequencies. As a consequence some circular ripple pattern emerges.

We point out that the performance variation as a function of x and y is about $\pm 12\%$, and if corrected for the circular ripple artifacts described above even lower. These numbers are encouraging, given that $\pm 10\%$ performance variation is deemed acceptable in commercial ultrasound transducers. As a further comparison: even calibrated hydrophones typically have a frequency dependent variation of $\pm 10 - 15\%$.

Implementation in manuscript:

P8. We have expanded the section describing the results of Figure 2c with information on the non-uniformity, so it now reads

The peak-to-peak efficiency variation over the total element area of 113 mm^2 was within $\pm 12\%$, except for two local spots with lower efficiencies. These spots result from small air bubbles on the surface of the transducer. Long-range random variations comes from process imperfections, yielding a spread in pillar geometry and/or poling efficiency. The circular ripple pattern in Fig 2c. is an artifact of the wavefield extrapolation algorithm that was used in combination with the finite transducer area.²⁶

Comment:

3. The formation of gas bubbles has been mentioned. How can this be prevented? Is the printing done under vacuum?

Reply:

As mentioned in the reply on question 1, the gas (air) bubbles are not present within the transducer itself, they reside on the surface of the device when placed upside down in a water bath. A brush can be used to wipe the surface clean of air bubbles, but in this experiment two micro-bubbles remained.

Implementation in manuscript:

See above.

Comment:

4. More information on the dependence of the cross-talk on the residual layer thickness and distance among pillars should be given.

Reply: The reviewer touches on an important point, and we are happy to provide more information on (the possible occurrence of) acoustic cross-talk as given in Supplementary Note 8.

Implementation in manuscript:

- P13. The element-to-element cross-talk was $\sim -31 \text{ dB}$. Its instantaneous nature implied it was mainly electrical, **indicating acoustic crosstalk is effectively reduced (see SI Note 8)**.
- Supplementary Note 8 is added.

Reviewer #2:

This manuscript presents a flexible ultrasonic device based on P(VDF-TrFE), featuring a microstructured thin film with a large aperture array. While certain distinctive aspects are highlighted by the authors—including the P(VDF-TrFE)-based transducer, pillar structure within each element, and the array's large area/aperture, to name a few—the novelty of the study appears constrained. These emphasized points have been documented in previous studies, and, as such, do not project a substantial level of innovation within this work. In addition, the application of monitoring blood pressure waveforms has been proficiently demonstrated by previous wearable ultrasound devices. The manuscript would also benefit from the inclusion of additional experimental results to lend greater credence to the conclusions drawn. Detailed comments are as follows.

Reply to this general comment:

We respectfully disagree with the statement of reviewer 2, and sincerely believe that the PillarWave technology represents a breakthrough. P(VDF-TrFE) has been disqualified by many researchers as piezoelectric transducer material, for incorrect reasonings that reviewer 2 is also mentioning below. We believe that the new and exciting *experimental results* shown in this manuscript for the first time prove that this view does not longer hold. The quality of the ultrasound image in Figure 4c speaks for itself plus important parameters such as the axial and lateral resolutions extracted from this image. Finally, it was realized with a material that is free of lead, and a simple technology that perfectly lends itself for ultrathin, flexible and large-area arrays.

Comment:

1. In Line 19, Page 2, the author mentioned that containing lead in the ultrasound transducer is a weakness. However, the transducer array is encapsulated well and PZT material would not touch to human tissue. People would not have chance to touch to lead. The reviewers are suggested to revise the content or provide other weakness of current ultrasound probe here. Similar to the sentence in Line 48, Page 3.

Reply:

Considering the toxicity of lead and its compounds, there is a general awareness for the development of environmental friendly lead-free materials. As long as there is no equivalent substitute for PZT, its use is temporarily allowed (the exemption protecting PZT in European RoHS is up for revision regularly). We therefore strongly feel that the use of lead in ultrasound transducers is to be avoided.

The reviewer points to the relative safety of lead when properly encapsulated, and although this point can be debated for a wearable device, we recognize that the main RoHS concerns originate from the synthesis of PZT and further preparation steps (in particular the high temperature sintering and calcination processes used and the micromachining, potentially causing environmental pollution, see f.i. <https://link.springer.com/article/10.1007/s10853-009-3643-0>). We therefore modified our original sentence, to avoid the suggestion that lead can leach out of the device during long-term skin-contact.

Implementation in manuscript:

p.3 The original sentence is now modified and reads 'Finally, the use of lead in these ultrasound transducers is to be avoided.'

Comment:

2. In Line 38, Page 3, the reference numbers of 6 to 10 have some typos. Please figure them out.

Reply: We thank the reviewer for pointing this out.

Implementation in manuscript:

The numbering is now correct.

Comment:

3. *P(VDF-TrFE) is often used as a receiver due to its low impedance and wide bandwidth. However, its poor transmission performance (having a much lower electromechanical coupling coefficient compared to piezoelectric ceramics) generally disqualifies it as a transmitter. Why did the authors choose to use P(VDF-TrFE) for the flexible ultrasound probe used for sensing and imaging? How did the authors overcome the intrinsic disadvantages of P(VDF-TrFE)?*

Reply:

The reviewer is correctly saying that P(VDF-TrFE) is an excellent receiver material and a relatively poor emitter. This may be made quantitative by the peak transmit transfer in kPa/V and the peak receive transfer in $\mu\text{V}/\text{Pa}$, which is for PillarWave 1.3 kPa/V (5.2 kPa/V with electrical tuning) and 147 $\mu\text{V}/\text{Pa}$, respectively. For imaging, pulse-echo ultrasound measurements are used, hence the product of said two parameters is the crucial figure of merit rather than the transmit efficiency alone. For PillarWave a pulse-echo efficiency of 0.19 was achieved. In the benchmark table in the SI we compare our results with those reported previously in academic manuscripts. The table below shows the comparison of the peak transmit, receive and composite pulse-echo transfer efficiencies with the best performing cMUT devices in industry [1]. As one can see the cMUT device has a 13 times higher peak transmit transfer, but a 13 times lower peak receive transfer compared to PillarWave, resulting in very similar peak pulse-echo efficiencies. Please note that the -6 dB bandwidths of both cMUT and PillarWave devices was at >100% similar. Hence, our conclusion is that the inherent pulse-echo performance of a PillarWave array is sufficient for imaging applications. This is also demonstrated by the good quality image obtained in Figure 4c, which was recorded real-time. (In fact the frame rate was limited by the real-time processing capabilities of the Verasonics and not the array.)

	Peak transmit transfer [kPa/V]	Peak receive transfer [$\mu\text{V}/\text{Pa}$]	Pulse-echo efficiency [V/V]
PillarWave	1.3	147	0.19
cMUT - best performing device from [1]	17	10.5	0.18

[1] Nenad Mihajlovic (nenad.mihajlovic@philips.com), 'A European MEMS Ultrasound Benchmark – Scientific White Paper', Nov 2021. This White Paper is based on the pan-European MEMS ultrasound transducer benchmark executed in the ECSEL Joint Undertaking project POSITION-2 (grant nr. Ecsel-783132-Position-II-2017-IA).

Having established that the P(VDF-TrFE) material is inherently a good transducer material the main issue for this material was crosstalk between neighbouring elements. We solved this by 'isolating' the elements mechanically by formation of the pillar structure.

Implementation in manuscript:

No changes were deemed necessary.

Comment:

4. In Line 54, Page 3, the authors mentioned the advantages of pillar structure in this study. However, those are not novel. In previous research, 1-3 composite transducers fabricated based on piezoelectric ceramics (such as PZT, PMN-PT) possess this structure, and the related arrays also have very small cross-talk (for instance: -70 dB in Science Advances, 2018, 4(3): eaar3979). Additionally, researchers have increased mechanical flexibility by filling the kerf of PZT pillars with silicone elastomer (IEEE Transactions on Industrial Electronics, 2019, 67(8): 6955-6962). Therefore, the authors are recommended to introduce some new advantages?

Reply:

We thank the reviewer for pointing us to work describing similar device structures. We will include these two references in our manuscript. Both manuscripts make use of ceramics that are micromachined and therefore materials and processes are fundamentally different from the PillarWave technology. Making pillar structures using embossed P(VDF-TrFE) is novel. We believe that the advantages of our approach are already well described elsewhere in the original manuscript.

Implementation in manuscript:

P3. References to the work of H. Hu et al. and T. Kim et al. are included so the new paragraph reads:

'... brings about two advantages. 1) It mechanically isolates neighboring elements. This strongly reduces acoustical crosstalk between neighboring acoustic elements, which is highly beneficial from a performance perspective.¹³ 2) It increases the mechanical flexibility of the overall array¹⁴ (see Supplementary Information (SI) note 5 ...'

.

Comment:

5. Please add more explanation in Line 65, Page 4 about why it is possible to enhance the acoustic bandwidth without adding a backing layer for P(VDF-TrFE)? If no need for backing layer, how is the ringing effect being addressed?

Reply:

The piezomaterial in a piezotransducer acts as a weakly damped harmonic oscillator. The Q factor (bandwidth) of the oscillator is determined by the effective losses in the oscillator. The losses can occur due to attenuation in the piezomaterial, or the leakage of acoustic energy towards the front (i.e.

tissue direction) or the back. Conventional piezomaterials typically have an acoustic impedance of 35 MRayl and tissue has an acoustic impedance close to that of water ~ 1.5 MRayl. Therefore, an airbacked slab of piezomaterial on top of tissue will have a relatively high Q factor, because the transmission coefficient from piezo to tissue will be low (amplitude transmission coefficient: $2 \cdot 1.5 / (1.5 + 35) = 0.082$). In traditional transducer design the effective transmission coefficient towards the tissue is increased by utilizing matching layers. The Q factor of the piezomaterial is further reduced by allowing some acoustic energy to escape into the backing, where it is absorbed. The use of matching layers and backings means that there is a trade-off between peak efficiency and bandwidth for conventional transducers.

In the case of PillarWave the impedance of the P(VDF-TrFE) is at 3.7 MRayl close to that of tissue, producing an amplitude transmission coefficient of $2 \cdot 1.5 / (1.5 + 3.7) = 0.58$. Compared to the transmission coefficient of 0.082 for a conventional bare piece of piezomaterial on top of tissue, the transmission coefficient for P(VDF-TrFE) on tissue is $>7x$ higher. In our case the bandwidth in reception is further enhanced by the transfer function of the electrical connection to the TIA and the TIA itself. Thus, the bandwidths typical for medical imaging can be reached without the need for matching layers or backings and the ringing of the transducer is very low.

Implementation in manuscript:

Page 4: With a value of 3.7 MRayl the measured acoustic impedance of the P(VDF-TrFE) pillars is close to that of human tissue. Hence, the transmission coefficient of a bare piece of regular piezomaterial (35 MRayl) into tissue is 0.082, whereas for bare P(VDF-TrFE) on tissue the transmission coefficient is 0.58. Therefore, the bandwidths typical for medical imaging can be reached without the need for matching layers or backings and the ringing of the transducer will be very low.

Comment:

6. In Line 95, Page 6, the authors mentioned corona poling was used for polarization. A potential issue with corona poling is the non-uniform polarization of the pillars. The uneven distribution of the corona charge could result in some sections of the material being polarized more strongly than others. This implies that a portion of the material might achieve a higher polarization intensity than other sections. Was this an issue in the study? Why not choose other methods that can achieve uniform polarization?

Reply:

We have studied both corona and contact poling for this work, and obtained better results with corona poling. Using contact poling we obtained very low device yield, and upon visual inspection saw evidence of local breakdown and electrical short formation between the top and bottom electrodes. One short effectively stops poling of the entire device.

In contrast, corona poling is known (<https://doi.org/10.1039/D2MA00559J> and <https://doi.org/10.3390/app11178108>.) to be much more forgiving in terms of breakdown particularly for large area, where now an electric breakdown occurs only locally and with limited damage to other areas in the transducer. This explains the higher device yield we could obtain with corona poling. In addition, this enabled us to apply higher electric fields, reaching higher values of d_{33} with corona poling compared to contact poling.

Implementation in manuscript:

P 6. We added “Both corona and contact poling were used to electrically polarize the structured piezoelectric layer. Corona poling gave less electrical breakdown and higher d_{33} values compared to contact poling. To promote uniformity over the large area, we employ a corona procedure where the sample constantly moves back and forward under a set of corona wires.”

Comment:

7. Please provide further explanation on how the thickness of P(VDF-TrFE), substrate, and encapsulation influences the resonance frequency? Additionally, factors influencing the resonance frequency of bulk PZT are not limited to thickness but also include the aspect ratio (thickness to width/diameter), as the aspect ratio can affect the vibration mode, thereby affecting the resonance frequency. Is this the case with P(VDF-TrFE)? Kindly elaborate.

Reply:

The fundamental resonance frequency is determined by the combined effects of the thickness of P(VDF-TrFE), substrate and encapsulation layer. One could create an effective medium with the compressional wave speed of one of the layers (the chosen layer) by increasing/decreasing the thicknesses of the other layers using the relative difference of the compressional wave speed of said other layer with the chosen layer.

We agree with the reviewer that the aspect ratio of the element affects the vibration mode and hence the resonance frequency. Typically for plate mode transducers (valid if the thickness = $< 1/10$ of the length/width of the plate) the velocity in the thickness (33) direction is given by $v_{33} = \sqrt{\frac{c_{33}^D}{\rho}}$, with c_{33}^D the stiffness coefficient measured in the thickness (33) direction and measured in closed circuit conditions and ρ the density. For rod mode transducers (valid when the thickness = > 10 of the length/width of the rod) the velocity in the thickness (33) direction is given by: $v_{33} = \sqrt{\frac{1}{s_{33}^D \rho}}$, with s_{33}^D the elastic compliance coefficient measured in the thickness (33) direction and measured in closed circuit conditions.

In our case the diameter of a hexagonal piezopillar is 40 μm and the thickness is 80 μm , so the vibration mode of the piezopillar is neither a pure thickness nor a pure rod mode. Hence, we've measured the sound speed and thickness of each device to calculate its resonance frequency. Later, we also measured the resonance frequency and the transmit and receive transfer functions as a function of frequency.

Implementation in manuscript:

No changes were deemed necessary.

Comment:

8. In Fig.2, some crucial properties, such as electromechanical coupling coefficient, sensitivity (insertion loss), dielectric loss, and the SNR of a single element, need to be provided. Although the literature has

related characterization results for P(VDF-TrFE), different fabrication methods can yield different performances, so actual characterization is necessary.

Reply:

We completely agree with the reviewer and point out that most of the parameters are provided in Supplementary information note 3). We have completed the parameter set, and refer to these crucial properties better in the revised manuscript. Please note that regarding the SNR we've provided the Noise Equivalent Pressure for the imaging array at page 13, as one needs for the noise level and/or SNR also the amplifiers designed for the element and the read-out system (in this case the Verasonics system).

Implementation in manuscript:

P 9: 'The obtained coupling factor (k_t) was 0.185, the relative permittivity (K_{33}^s) was 6.5-0.0088*frequency (MHz), the dielectric loss tangent ($\tan(\delta_e)$) was 0.075+0.04*frequency (MHz), and the mechanical loss tangent ($\tan(\delta_m)$) was found to be 0.125. The pulse-echo insertion loss was calculated to be -20.4 dB. For more information see Supplementary information note 3.'

Comment:

9. In Line 177 Page 9, the author mentioned the experiment was done in the oil because the acoustic impedance of the oil was similar to that of water. How about the acoustic attenuation of oil? If the acoustic attenuation of oil and water is different, does this influence the result of characterization? In addition, why can the experiments in Fig.3 be conducted in water, but not the ones in Fig. 2?

Reply:

The attenuation of oil was indeed higher than the attenuation in water. The comment in line 177 page 9 refers to the electrical impedance measurement results of Fig. 2d-e. The shape of the electrical impedance curves are dominated by the high amplitude transmission coefficient (0.58) from P(VDF-TrFE) to water/oil, due to the low acoustic impedance of P(VDF-TrFE) (3.7 MRayl) relative to the acoustic impedance of water/oil (1.5 MRayl). Moreover, the electrical impedance curves were affected by the dielectric and mechanical losses. The acoustic attenuation in the oil does add some damping, but it is a very minor effect.

Fig. 2d-e show the modeled and measured magnitude and phase of the electrical impedance as a function of frequency for the condition where the single element transducer was in air or the front face was in contact with oil. Here, electrically isolating oil was used instead of water, since the electrical isolation of this early single element prototype was suboptimal.

The 128 element linear array prototype transducer used in Fig. 3 featured improved electrical isolation, since the uniformity of the layer thickness touching the water was improved. This meant the top electrode (the electrode closest to the water) was physically isolated from the water.

Implementation in manuscript:

No changes were deemed necessary.

Comment:

10. In Fig. 2d, why are the resonant and anti-resonant frequencies not distinct in oil (similar to Fig. 2e about the phase angle)? If the resonant frequency and piezoelectric properties would change under a load, then would the frequency and other properties also change when the device is tested on a person?

Reply:

The resonant and anti-resonant frequencies are not very distinct, since:

- The acoustic impedance of the P(VDF-TrFE) is at 3.7 MRayl close to that of oil/water (1.5 MRayl). That means the amplitude transmission coefficient for the acoustic wave traveling from P(VDF-TrFE) to oil/water is 0.58. Hence, the P(VDF-TrFE) will show a large frequency bandwidth and a low Q-factor.
- The P(VDF-TrFE) piezopolymer features relatively high dielectric and mechanical loss tangents.
- The coupling factor (k_t) is at 0.185 relatively low compared to regular rigid piezomaterials ($k_t = \sim 0.5$)

Because the oil used, had an acoustic impedance very similar to that of tissue/water (1.5 MRayl) no differences in center frequency or transfer functions were observed.

Implementation in manuscript:

On P 9, we added: '..., hence no significant difference in center frequency or transfer function was expected and observed'

Comment:

11. In Fig. 3c, because the element is soft, the acoustic field distribution emitted by the element could vary when the probe is placed on objects with different curvatures, resulting in different pulse-echoes intensities. Therefore, the authors are advised to perform more comprehensive characterizations demonstrating pulse-echo and SNR under different curvatures. The intensity of the reflected pulse-echo determines the range of curvature for practical applications.

Reply:

The aim of Fig. 3 is to show that the PillarWave transducer can be flexed to a small radius and still produce pulse-echo signals with good SNR. We agree that the curving of the array will deform the wavefield. Limited hydrophone experiments have shown that the effect of the curvature on the wavefield can be explained by the 3D shape of the aperture. We agree with the reviewer that a design of an optimized PillarWave EUS device will require a dedicated optimized transducer design taking into account the effect of the array curvature on the wavefield.

We point out that in this article we provide quantitative performance numbers for a PillarWave based 128 element linear array transducer intended for imaging of the carotid (see Figure 4) and for a PillarWave based large aperture array transducer (91.2 x 14 mm²) for blood pressure monitoring applications.

Implementation in manuscript:

On P 11, we added: “Preliminary experiments show an effect of the curvature on the wavefield, and a design of an optimized EUS device will need to take into account this effect.”

Comment:

12. The authors tested the pulse-echo while wrapped on an EUS probe in Fig.3, and characterized the spatial resolution of imaging in Fig.4, yet the final application (Fig.5) did not involve either the EUS probe or imaging, rendering the manuscript logically inconsistent. From reviewer’s understanding, the characterization should serve the final application to some extent, to render the characterization meaningful.

Reply:

It is not the intent of the authors to work towards a single application in this manuscript. Rather, we believe that the unique properties of the PillarWave technology can be utilized in various ultrasound applications. Each demonstrator is meant to illustrate one specific aspect. The EUS probe shows that our transducer can be flexed to a very small radius. The high-resolution imaging shows that it is possible to obtain high quality images, with good spatial resolution. The blood pressure sensor illustrates the advantage of using very large element sizes.

We have tried to clarify our intentions even better in the revised manuscript.

Implementation in manuscript:

On P 20 we rephrased, so it now reads: “The potential of the technology is demonstrated in three applications, each showing a specific unique advantage of the PillarWave™ technology – an endoscopic ultrasound (EUS) transducer that remains functional while mechanically curved over a small radius of 3 mm, b-mode imaging of the carotid showing good spatial resolution, and a blood pressure sensor that illustrates the potential to scale to large-area. We point out that in all cases the same fabrication process was used, using a single P(VDF-TrFE) pillar geometry.”

Comment:

13. A compelling demonstration/application should address a genuine, existing problem. The application in this manuscript, blood pressure waveform monitoring, has been achieved by previous conformal ultrasound devices. In addition, the test results in this study do not exhibit superior quality comparing with previous results. Moreover, the blood pressure waveform can be tested with just one element and does not require a large aperture array. Therefore, the authors are suggested to design another application and provide comprehensive data to showcase the unique advantages of a large aperture array that addresses a new clinical problem.

Reply:

We thank the reviewer for the suggestion. As underlined in the previous reply, the purpose of this manuscript is to describe a new transducer technology that offers a unique combination of advantages that can be exploited in various ways, as shown in three compelling demonstrations/applications each of which representing relevant clinical cases.

Implementation in manuscript:

See comment above.

REVIEWER COMMENTS

Reviewer #1 (Remarks to the Author):

The authors have exhaustively replied to my questions.

Reviewer #2 (Remarks to the Author):

The authors have made certain modifications based on the reviewer's comments, which has improved the overall quality of the manuscript, especially in the explanation of some principles (e.g., reply to comment 10). However, there is a need for an increased amount of experimental data to better demonstrate the device's properties and functionalities. Moreover, looking at articles published in Nature Communications on wearable medical devices, they not only feature innovative device fabrication techniques, structures, et al. but also extensively showcase innovative applications of the devices; both aspects are indispensable. Therefore, more improvement in this work may be needed. Below are some detailed comments.

1. In reply to comment 3, although PillarWave and CMUT share similar pulse-echo efficiency and bandwidth, CMUT operates at a higher bias voltage, ranging from tens to hundreds of volts, compared to the voltage required to drive P(VDF-TrFE), which is only a few to tens of volts. This higher bias voltage endows CMUT with greater sensitivity, a key factor in its ability to produce good imaging. In addition, in Fig. 4c, the reflectors and surrounding materials exhibit significant acoustic impedance differences, leading to a substantial reflection of ultrasound waves at the reflectors. However, evaluating the imaging capabilities of PillarWave solely based on this test is not comprehensive. The authors should conduct additional experiments, using both PillarWave and traditional ultrasound probes (either PZT-based or CMUT) to scan a phantom or human tissue with smaller acoustic impedance differences. Images at various depths should also be obtained. It will thoroughly help analyze the imaging capabilities of PillarWave.

2. In reply to comment 9, it is suggested that the experiments in oil from Fig. 2 be redone in water, and the data from water be incorporated into Fig. 2d and 2e.

Response to the comments of the reviewers on the revised manuscript “Flexible large-area ultrasound arrays for medical applications using embossed polymer structures” by Van Neer *et al.* submitted to *Nature Communication*.

We thank the reviewers for their favorable comments and their thoughtful comments and suggestions to improve the manuscript. Below we repeat these comments and then provide our response and outline the changes made in the manuscript.

Reviewer #1 (Remarks to the Author):

Comment:

The authors have exhaustively replied to my questions.

Reply:

We thank the reviewer for his/her favorable comments and thoughtful comments and suggestions to improve the manuscript.

Reviewer #2 (Remarks to the Author):

Comment:

The authors have made certain modifications based on the reviewer's comments, which has improved the overall quality of the manuscript, especially in the explanation of some principles (e.g., reply to comment 10). However, there is a need for an increased amount of experimental data to better demonstrate the device's properties and functionalities. Moreover, looking at articles published in Nature Communications on wearable medical devices, they not only feature innovative device fabrication techniques, structures, et al. but also extensively showcase innovative applications of the devices; both aspects are indispensable. Therefore, more improvement in this work may be needed. Below are some detailed comments.

Reply to the overall comment:

We agree with the opinion of reviewer #2 that Nature Communications articles should feature ‘innovative device fabrication techniques, structures, etc as well as innovative applications of the devices’. We believe that our article fulfils this criterium.

Comment:

1. *In reply to comment 3 [of the revised manuscript], although PillarWave and CMUT share similar pulse-echo efficiency and bandwidth, CMUT operates at a higher bias voltage, ranging from tens to hundreds of volts, compared to the voltage required to drive P(VDF-TrFE), which is only a few to tens of volts. This higher bias voltage endows CMUT with greater sensitivity, a key factor in its ability to produce good imaging.*

Reply:

Reviewer 2 correctly states that CMUTs typically use higher voltages to improve their image producing quality. Yet, high voltages are disadvantageous for wearable applications (power consumption, higher cost of driving electronics, safety aspects). Demonstrating similar pulse-echo efficiency and bandwidth at lower voltages works therefore in favor of our approach.

Implementation in manuscript:

No changes were deemed necessary.

Comment:

In addition, in Fig. 4c, the reflectors and surrounding materials exhibit significant acoustic impedance differences, leading to a substantial reflection of ultrasound waves at the reflectors. However, evaluating the imaging capabilities of PillarWave solely based on this test is not comprehensive. The authors should conduct additional experiments, using both PillarWave and traditional ultrasound probes (either PZT-based or CMUT) to scan a phantom or human tissue with smaller acoustic impedance differences. Images at various depths should also be obtained. It will thoroughly help analyze the imaging capabilities of PillarWave.

Reply:

We respectfully disagree with the reviewer. The imaging quality of PillarWave in Fig 4c was assessed using a commercially available tissue phantom. Parameters such as axial resolution, lateral resolution, frequency bandwidth, etc were derived using international standardized methods IEC 61391.1:2006 and IEC 61391-2:2006. This, together with the extended benchmark table in the supporting information, will help to thoroughly understand the imaging capabilities of our arrays.

The phantom we used is one of the standards for ultrasound quality assurance. It has reflectors and anechoic chambers that are generally accepted to be representative for medical ultrasound imaging. For more details: [Multi-Purpose Multi-Tissue Ultrasound Phantom - CIRS \(cirsinc.com\)](https://www.cirsinc.com). We therefore disagree with the statement of the reviewer that the contrast and high quality of the image in Fig 4c is good due to unrealistic acoustic impedance differences of the phantom (if we interpret the comment of the reviewer correctly).

We believe that the use of a commercial tissue phantom is a strong point of the paper, and allows benchmarking different ultrasound technologies. We therefore decided to modify the manuscript accordingly.

Implementation in manuscript:

On P 14, we modified the text of the manuscript so it now reads: 'Fig. 4c shows a typical ultrasonic B-mode image obtained using a tissue mimicking phantom (040GSE, CIRS, Norfolk, Virginia, USA). Such phantoms provide an invaluable approach for objective, quantitative evaluation of image quality characteristics. The image was obtained using the 128-linear array on so-called plane wave compounding (of 9 plane wave transmissions) in combination with delay-and-sum beamforming in reception. The reflections of the nylon wires are clearly visible with smooth and sharp point-spread-functions (PSFs). The imaging capabilities of PillarWave in Fig 4c were quantified using international standardized methods IEC 61391.1:2006 and IEC 61391-2:2006. ...'

Comment:

2. In reply to comment 9, it is suggested that the experiments in oil from Fig. 2 be redone in water, and the data from water be incorporated into Fig. 2d and 2e.

Reply:

Repeating the experiments in water was deemed unnecessary, as we explained in our previous reply. The argumentation why oil was used and deemed relevant is now incorporated in the text of the manuscript, for maximum transparency.

Implementation in manuscript:

On P 9, we modified the text of the manuscript so it now reads: 'Here, electrically isolating oil was used instead of water, since the electrical isolation of this early prototype was suboptimal. The acoustic impedance of the oil was similar to that of water. The shape of the electrical impedance curves are dominated by the high amplitude transmission coefficient (0.58) from P(VDF-TrFE) to water/oil, due to the low acoustic impedance of P(VDF-TrFE) (3.7 MRayl) relative to the acoustic impedance of water/oil (1.5 MRayl) and therefore center frequency and transfer function of the measurements in oil are considered representative. The higher acoustic attenuation in the oil does add some damping, but it is a very minor effect.'